plant science/biochemistry

curcumin, adipocyte browning, mitochondria, PPARγ, antiobesity

**Authors for correspondence:**
Guanjian Jiang
e-mail: guangjianjiang@sina.com
Sihua Gao
e-mail: gaosh@bucm.edu.cn

[†]These authors contributed equally to this study.

# Curcumin improves adipocytes browning and mitochondrial function in 3T3-L1 cells and obese rodent model

Dandan Zhao[1,†], Yanyun Pan[1,†], Na Yu[2], Ying Bai[1], Rufeng Ma[1], Fangfang Mo[1], Jiacheng Zuo[1], Beibei Chen[1], Qiangqiang Jia[1], Dongwei Zhang[1], Jiaxian Liu[3], Guanjian Jiang[1] and Sihua Gao[1]

[1]Traditional Chinese Medicine School, Beijing University of Chinese Medicine, Beijing 100029, People's Republic of China
[2]Educational Office, Beijing Tian Tan Hospital, Capital Medical University, Beijing 100050, People's Republic of China
[3]Leonard Davis School of Gerontology, University of Southern California, Los Angeles, CA 90007, USA

DZ, 0000-0002-0757-984X; YB, 0000-0002-7569-0099; GJ, 0000-0001-8394-5904; SG, 0000-0002-0332-3767

Accumulating evidence suggests that mitochondrial dysfunction and adipocyte differentiation promote lipid accumulation in the development of obesity and diabetes. Curcumin is an active ingredient extracted from *Curcuma longa* that has been shown to exhibit antioxidant and anti-inflammatory potency in metabolic disorders. However, the underlying mechanisms of curcumin in adipocytes remain largely unexplored. We studied the effects of curcumin on adipogenic differentiation and mitochondrial oxygen consumption and analysed the possible mechanisms. 3T3-L1 preadipocytes were used to assess the effect of curcumin on differentiation of adipocytes. The Mito Stress Test measured by Seahorse XF Analyzer was applied to investigate the effect of curcumin on mitochondrial oxygen consumption in 3T3-L1 adipocytes. The effect of curcumin on the morphology of both white and brown adipose tissue (WAT and BAT) was evaluated in a high-fat diet-induced obese mice model. We found that curcumin dose-dependently (10, 20 and 35 μM) induced adipogenic differentiation and the intracellular fat droplet accumulation. Additionally, 10 μM curcumin remarkably enhanced mature adipocyte mitochondrial respiratory function, specifically, accelerating basic mitochondrial respiration, ATP

production and uncoupling capacity via the regulation of peroxisome proliferator-activated receptor γ (PPARγ) ($p < 0.01$). Curcumin administration also attenuated the morphological changes in adipose tissues in high-fat diet-induced obese mice. Moreover, curcumin markedly increased the mRNA and protein expressions of mitochondrial uncoupling protein 1 (UCP1), PPARγ, peroxisome proliferator-activated receptor-γ coactivator-1α (PGC-1α) and PR domain protein 16 (PRDM16) *in vivo* and *in vitro*. Collectively, the results demonstrate that curcumin promotes the adipogenic differentiation of preadipocytes and mitochondrial oxygen consumption in 3T3-L1 mature adipocytes by regulating UCP1, PRDM16, PPARγ and PGC-1α expression.

# 1. Introduction

In recent decades, the obese population has skyrocketed worldwide, and insulin resistance due to increased adipose tissue mass, predominantly in viscera, has been recognized as a pivotal factor underlying the equivalent increase in type 2 diabetes mellitus. Countermeasures to control the enlargement of white adipose tissue (WAT) and induction of white adipocyte browning may offer a new solution in fighting obesity and its associated diseases [1,2]. Methods to prevent the enlargement of adipose tissues (i.e. preventing fat accumulation) and induce the browning of white adipocytes (promoting energy expenditure) are the focus of current research on obesity and related diseases.

Mitochondria are considered the cellular 'powerhouse' since they provide the site and essential materials for various metabolic processes during which ATP is generated from the chemical energy contained in three major energy sources [3]. Usually, high levels of saturated fatty acid bring about mitochondrial functional abnormality and inflammation during the progress of insulin resistance in obese individuals [4,5]. Numerous studies have demonstrated that the amount and oxidative function of mitochondria altered notably in adipogenesis and the development of metabolic disorders [6,7]. In addition, mitochondria are necessary in the differentiation of adipocytes, in which a lot of ATP are needed when adipocytes are activated with metabolic function. Therefore, compounds that improve mitochondrial oxygen consumption may exhibit therapeutic efficacy in the prevention and control of many diseases due to energy metabolism disorders, such as obesity and diabetes.

Curcumin (1,7-bis(4-hydroxy-3-methoxyphenyl)-1,6-heptadien-3,5-dione) is an active ingredient extracted from turmeric (*Curcuma longa*), which is not only a traditional Chinese herb but also a dietary spice. Accumulating evidence has demonstrated that it could prevent and treat a wide range of diseases by its antioxidant and anti-inflammatory properties in preclinical trials, for instance, arthritis, cancer, neurodegenerative disorders, metabolic syndrome, cardiovascular diseases, diabetes and so on [8,9], and the same potential has been validated by many clinical trials [10]. Specifically, curcumin could regulate various molecular targets, such as nuclear factor κB (NF-κB), tumor necrosis factor-α (TNFα), and monocyte chemoattractant protein-1 (MCP-1), which might contribute to its therapeutic potential in obesity and diabetes [11]. Recently, curcumin was demonstrated to regulate inflammation and restore redox homeostasis among postpubescent overweight and obese girl adolescents [12]. In addition, we previously showed that curcumin improves glycolipid metabolism in 3T3-L1 cells and experimental mice with diabetes via the peroxisome proliferator-activated receptor γ (PPARγ) signalling pathway [13]. However, whether the effects of curcumin on regulating metabolism are actualized via mitochondrial function remains to be determined. Therefore, the present study aimed to elucidate the influence of curcumin on mitochondrial function and adipocyte browning and the molecular mechanisms.

# 2. Material and methods

## 2.1. Reagents

Curcumin (the chemical structure is shown in figure 1) with a purity of 98% was purchased from Sigma (cas.#: 458-37-7, St. Louis, MO, USA). Dulbecco's Modified Eagle's Medium (DMEM), penicillin-streptomycin, together with 0.25% trypsin-EDTA were obtained from Gibco (Rochester, NY, USA). Fetal bovine serum (FBS) was purchased from Tian Hang Bio-company (Hangzhou, China). Insulin, dexamethasone (DEX) and 3-isobutyl-1-methylxanthine (IBMX) were bought from Sigma. Oil Red O was obtained from Solarbio Biocompany (Beijing, China). The XF Cell Mito Stress Test kit was

**Figure 1.** Chemical structure of curcumin.

purchased from Seahorse Bioscience (Billericay, MA, USA). GW9662 was purchased from Selleck (cat.# S2915, Shanghai, China). Antibodies for uncoupling protein 1 (UCP1), PPARγ, peroxisome proliferator-activated receptor-γ coactivator-1α (PGC-1α) and GAPDH were purchased from Protein Tech Group (cat.# 23673-1-AP, 20658-1-AP, 16643-1-AP and 10494-1-AP, Rosemont, IL, USA). The antibody against PR domain protein 16 (PRDM16) was bought from Abcam, Inc. (cat.# ab106410, Cambridge, MA, USA). Anti-rabbit lgG-HRP was purchased from Cell Signaling Technology, Inc. (Beverly, MA, USA). Unless otherwise specified, all other reagents were obtained from Beijing Sinopharm Chemical Group (Beijing, China).

## 2.2. Culture and differentiation of adipocytes

3T3-L1 preadipocytes were acquired from the CAMS & PUMC Cell Center and were then cultured in DMEM containing 10% FBS and 1% penicillin-streptomycin at the temperature of 37°C and 5% $CO_2$ with saturated humidity. Cells were seeded in 24-well culture plates, and then induced to differentiation by incubating in DMEM supplemented with MDI (0.5 mM IBMX, 1.0 µM DEX and 10 µg ml$^{-1}$ insulin) for 2 days and further culturing in DMEM with 10 µg ml$^{-1}$ insulin for an additional two days. Subsequently, the DMEM was renewed every 48 h until the cells were fully differentiated.

Oil Red O was adopted to determine differentiated intensity of 3T3-L1 cells, which has been used in our previous studies [14]. And differentiated mature 3T3-L1 cells were applied in the following experiments.

## 2.3. Cell counting kit-8 assay

Curcumin was dissolved in dimethyl sulfoxide (DMSO) and prepared as a 10 mM stock solution. Then the stock was diluted in the medium to the indicated concentration for the experiment. The differentiated 3T3-L1 adipocytes were cultured with the medium supplemented with curcumin of several concentrations for 24 h, and then the relative cell viability was calculated as the percentage of untreated cells applying the CCK-8 assay. The doses of curcumin used in this study were determined based on previous investigations [15,16]. Briefly, 10 µl of CCK8 were incubated with the cells for another 2 h before the optical density (OD) values were read at 450 nm using a microplate reader.

## 2.4. Oil Red O staining

Oil Red O staining was applied to measure the lipid accumulation intracellularly. Briefly, after the cells were incubated with or without curcumin (0, 10, 20 and 35 µM) for 24 h, the culture medium was removed, and the cells were rinsed in phosphate buffer saline (PBS) for 3 × 5 min and then fixed in 4% formaldehyde for half an hour. Next, a dilute solution of Oil Red O (3:2) was filtered using a 0.45 µm filter followed by incubating the fixed cells for 1 h. All above procedures were carried out at room temperature. Then, the stained lipid droplets in cytoplasm were visualized by microscopy. The Oil Red O-stained 3T3-L1 cells were dissolved in 200 µL of isopropyl alcohol and transferred to a 96-well microplate. The absorbance of the dissolved Oil Red O was detected at 510 nm by a microplate reader.

## 2.5. Oxygen consumption rate measurements

The oxygen consumption rate (OCR) in 3T3-L1 adipocytes was measured to evaluate oxidative phosphorylation using the Seahorse Biosciences XF Extracellular Flux Analyzer (Seahorse Bioscience, North Billerica, MA, USA) following the manufacturer's operating instructions. In brief, 100 µl of 3T3-L1 cell suspension (2 × 10$^4$ cells) were seeded in each well of a 24-well XF Cell Culture Microplate.

When the cells were differentiated into mature adipocytes, curcumin was added at various concentrations (10, 20 and 35 µM) for 48 h. Then the growth medium was exchanged with XF basic medium (pH = 7.4), which contained $4.5\,g\,l^{-1}$ glucose, 4.0 mM glutamine and 1.0 mM sodium pyruvate. OCR measurements were performed following the sequential addition of 2.0 µM oligomycin, 1.0 µM FCCP and 1.0 µM rotenone/antimycin. All above experimental procedures were carried out at 37°C. The following mitochondrial parameters were determined: basal respiration, ATP production (basal respiration subtracted oligomycin-inhibited respiration), maximal respiratory capacity (maximal uncoupled respiration subtracted non-mitochondrial respiration) and spare respiratory capacity (maximal uncoupled respiration subtracted basal respiration). The results are expressed in pmol $(O_2).min^{-1}$ and normalized to the cell number.

## 2.6. Animals

Thirty-six male C57BL/6 J mice (about 20 g, six-week old) were purchased from Beijing HFK Bioscience Co., Ltd. (Beijing, China; permit number SCXK 2016-0038) and divided into two groups that received two different diets. Ten mice in the control group were fed a standard chow diet comprising 10% kcal (MD12031) of fat, while the remaining mice in the other group were given a high-fat diet containing 60% kcal (MD12033) of fat for 12 weeks. Twenty mice in the high-fat diet group were confirmed as obese as their body weight was 20% greater than their peers. The food was purchased from Mediscience Co., Ltd. (Jingsu, China), and the diet compositions of MD12031 and MD12033 are listed in electronic supplementary material, 1 (S1). Next, curcumin (Cur, $n = 10$) was administered at 50 mg $kg^{-1}\,d^{-1}$ by oral gavage to the obese mice for an additional eight weeks. The same volume of saline was then given by gavage to the obese (diet-induced obesity (DIO), $n = 10$) and normal mice (Normal, $n = 10$). During this period, all the mice were maintained on the original diet. After eight weeks of intervention, the mice were sacrificed under pentobarbital anaesthesia (60 mg $kg^{-1}$) after fasting for 12 h. Adipose tissue was removed and stored at −80°C for histological and molecular biological analysis. Six samples were selected randomly from each group for further analysis of histological evaluation, and gene expression and protein expressions. And for each type of molecular biology analysis, the adipose tissues were from the same mice.

## 2.7. Histological analysis

After careful excision with a scalpel blade, subcutaneous WAT and scapular brown adipose tissue (BAT) samples were fixed in 10% paraformaldehyde and then dehydrated in a gradient ethanol series (50%, 70%, 80%, 90%, 95% and 100%). The samples were defatted in xylene, embedded in paraffin, dyed with haematoxylin-eosin and photographed under a microscope (Olympus, Japan). The sizes of WAT and BAT cells were measured by counting 50 cells using IMAGEJ software.

## 2.8. Western blot analysis

Adipose tissues and cells were lysed using RIPA buffer with a protease inhibitor cocktail, 20 mM Tris, 1 mM EDTA, 140 mM NaCl, 1% Nonidet P-40 (NP-40), 50 U of aprotinin/ml, 1 mM $Na_3VO_4$, 1 mM PMSF and 10 mM NaF (pH 7.5), and the proteins were quantified applying the Micro BCA™ Protein Assay Reagent Kit. Protein separation was performed using SDS-PAGE gel electrophoresis with 40 mg protein per well. After electrophoresis, the proteins were subsequently transferred onto a PVDF membrane, which were next incubated with UCP1, PRDM16, PGC-1α and PPARγ antibodies (1 : 1000 dilution) overnight at 4°C. The membrane was then incubated with a corresponding secondary antibody (1 : 2000). Blots were generated using an ECL kit under the guidance of the manufacturer's instructions. The greyscale of protein bands was analysed by IMAGEJ software. GAPDH was used as a loading control.

## 2.9. Real-time PCR

Total RNA was isolated from adipose tissue and 3T3-L1 cells applying TRIzol reagent in accordance with the operating manual provided by the manufacturer. Reverse transcriptases (RTs) from the isolated RNA into cDNA were then carried out following by quantitative real-time PCR which was performed in accordance with the protocol recommended by the manufacturer (ABI 7500, USA). The primers used in the current study were listed in table 1. ARBP was used as a control for normalization. Eventually,

**Table 1.** Sequence of specific primers used in real-time polymerase chain reaction.

| genes | primer sequence (5′→3′) | |
| --- | --- | --- |
| | forward primer | reverse primer |
| UCP1 | GGGCCCTTGTAAACAACAAA | GTCGGTCCTTCCTTGCACTT |
| PRDM16 | GACATTCCAATCCCACCAGA | CACCTCTGTATCCGTGTGTA |
| PPARγ | TCAGCTCTGTGGACCTCTCC | ACCCTTGCATCCTTCCAGCA |
| PGC-1α | CCCTGCCATTGTTAAGACC | TGCTGCTGTTCCTGCTCCT |
| ARBP | TTTGGGCATCACCACGAAAA | GGACACCCTCCAGAATTTTC |

data were analysed using the $2^{-\Delta\Delta CT}$ relative quantification method according to the manufacturer's instructions.

## 2.10. Statistical analysis

SPSS software (v. 20.0) was applied for statistical analyses. The results are presented as the means ± standard errors (Mean ± s.e.m.). One-way analysis of variance was applied for comparisons among groups, followed by the LSD or SNK test for comparisons between two groups. The difference was considered significant if the $p$-value is less than 0.05.

# 3. Results

## 3.1. Effects of curcumin on the 3T3-L1 cells viability and differentiation

As shown in figure 2$a$, curcumin exhibited a subtle influence on preadipocyte viability at 10, 20 and 35 μM compared to that in the control group ($p > 0.05$). However, the viability of the preadipocytes decreased by approximately 55.0% and 74.7% compared to that in the control group upon incubation with 50 μM and 75 μM curcumin, respectively ($p < 0.01$). The results indicated that high concentration of curcumin may have cell toxicity.

Next, the 3T3-L1 cells were incubated with different concentrations (0, 10, 20 and 35 μM) of curcumin for 8 days to induce differentiation. The results of Oil Red O staining revealed that different concentrations of curcumin accelerated adipocyte differentiation and increased the numbers of mature adipocytes (figure 2$b$). In addition, curcumin could increase the triglyceride level in differentiated 3T3-L1 cells (figure 2$c$). These results suggest that curcumin could influence lipid profiles though promotion of adipocyte differentiation.

## 3.2. Effects of curcumin on mitochondrial function in 3T3-L1 cells

As shown in figure 3$a$, a typical bioenergetic profile, involved in a four-step analysis: (1) basal OCR, 3T3-L1 cells were incubated in normal medium; (2) ATP synthesis turnover, oligomycin (2.0 mM) was supplemented to the medium to inhibit ATP synthase; (3) maximal mitochondrial respiratory capacity, cells were motivated with FCCP (1.0 mM); and (4) non-mitochondrial respiration, rotenone (1.0 mM) was introduced to inhibit complex I.

As shown in figure 3$b$, the OCR of curcumin-treated (10 μM) adipocytes remained higher than that of adipocytes in the control group. Notably, different doses of curcumin (20 and 35 μM) exerted opposite effects on the mitochondrial respiratory function of mature adipocytes. Specifically, the basal mitochondrial respiration in the curcumin-treated (10 μM) group was elevated by 16.5% compared to that of the control, but 20 and 35 μM curcumin reduced basal mitochondrial respiration by 13.6% and 49.5%, respectively (figure 3$c$, $p < 0.05$). This difference resulted in a 27.9% increase and 22.6% and 37.5% decreases in the ATP production of cells treated with curcumin at 10, 20 and 35 μM, respectively, compared to that in control cells (figure 3$d$, $p < 0.05$). Moreover, interventions with different doses of curcumin altered the maximal mitochondrial respiratory capacity of adipocytes. As shown in figure 3$e$, 10 μM curcumin enhanced the maximal mitochondrial respiratory capacity by

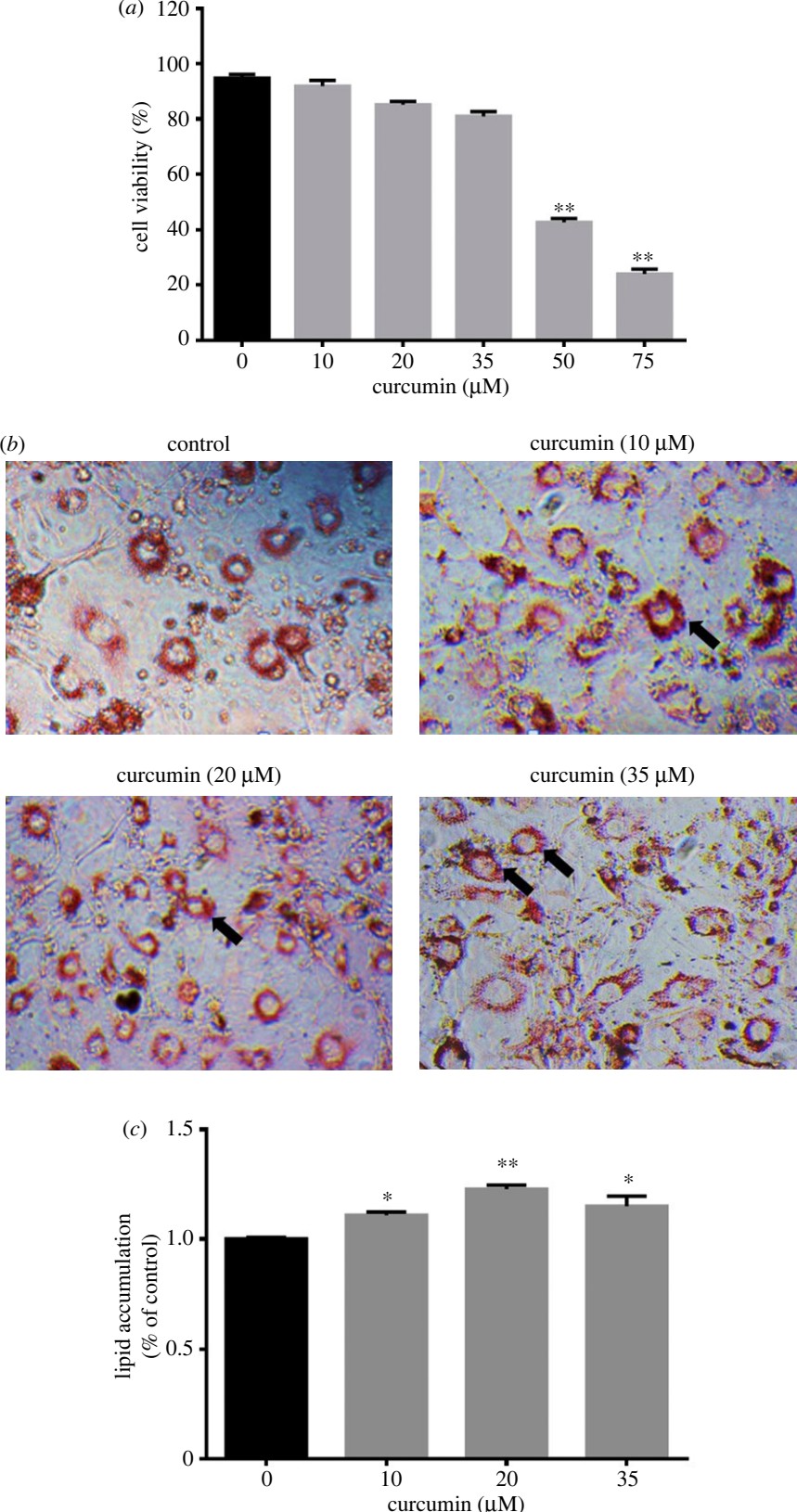

**Figure 2.** Curcumin influenced the 3T3-L1 cells viability and differentiation. (*a*) Cells relative viability was calculated by cell count kit-8 reagent. (*b*) Curcumin-treated 3T3-L1 cells stained with Oil Red O. The arrows indicate the typical adipocytes. (*c*) Lipid accumulation of curcumin-treated 3T3-L1 cells. Cells were differentiated into mature adipocytes for 8 days in DMEM with or without curcumin (0, 10, 20 and 35 µM). Intracellular lipid droplets were stained by Oil Red O and photographed under a microscope (magnification: 400×). Data are presented as the mean ± s.e.m. from three repeated experiments. $^{**}p < 0.01$ compared with the untreated control.

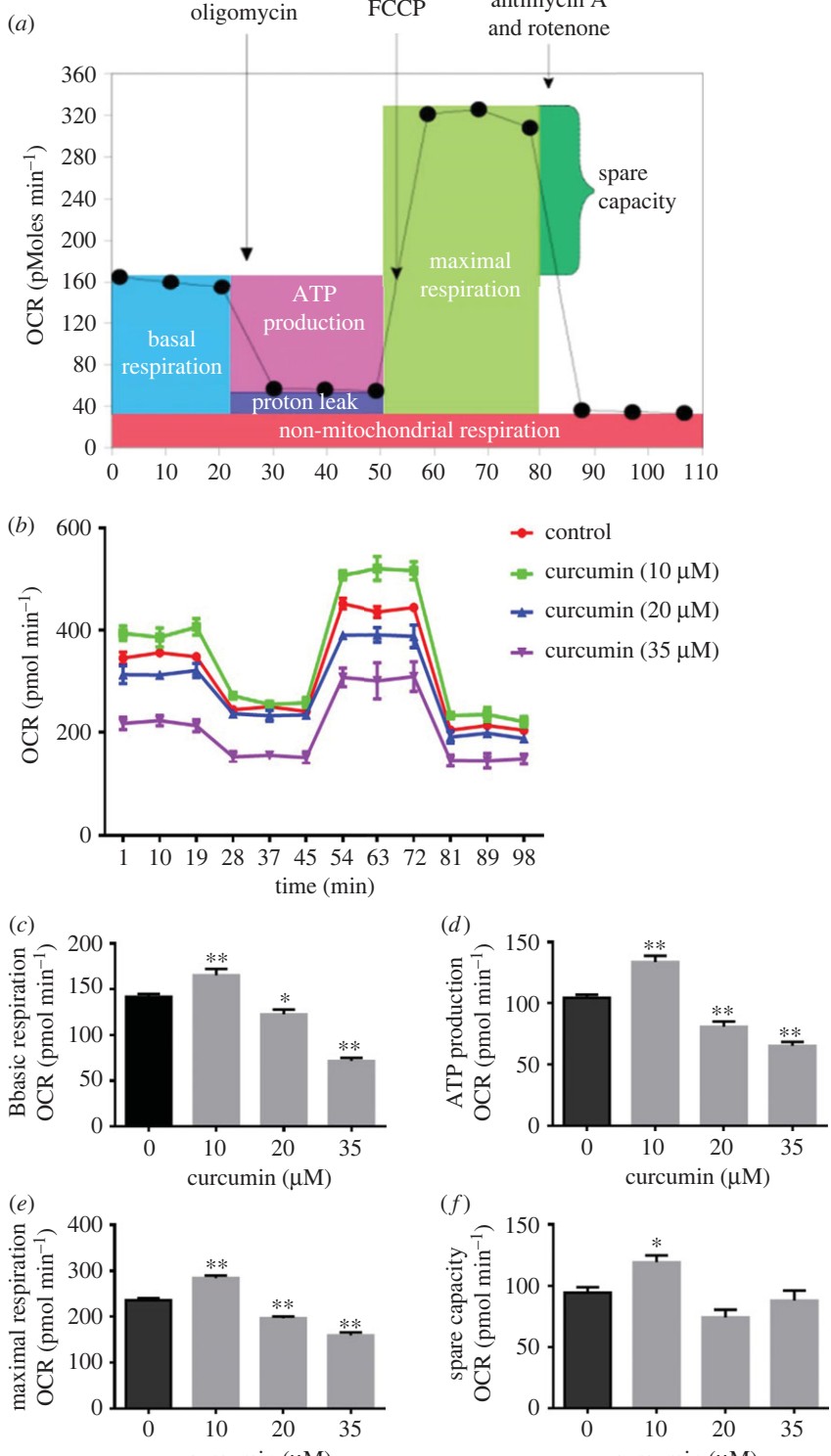

**Figure 3.** OCR in curcumin-treated 3T3-L1 mature adipocytes. (*a*) A typical bioenergetics profile involving a four-step analysis. (*b*) OCR curves obtained from 3T3-L1 mature adipocytes treated with different concentrations of curcumin. (*c*) Basal respiration. (*d*) ATP production. (*e*) Maximum respiration. (*f*) Spare capacity. Data are presented as the mean ± s.e.m. from three repeated experiments. $^*p < 0.05$, $^{**}p < 0.01$ compared with the untreated control.

20.5%. However, the maximal mitochondrial respiratory capacity was reduced by 16.7% and 32.4% in cells treated with 20 μM and 35 μM curcumin, respectively, compared to that of the control ($p < 0.05$). There was no significant difference in the spare capacities of cells treated with 20 and 35 μM curcumin ($p > 0.05$), but the spare capacity of the 10 μM curcumin treatment group was increased by 26.4%

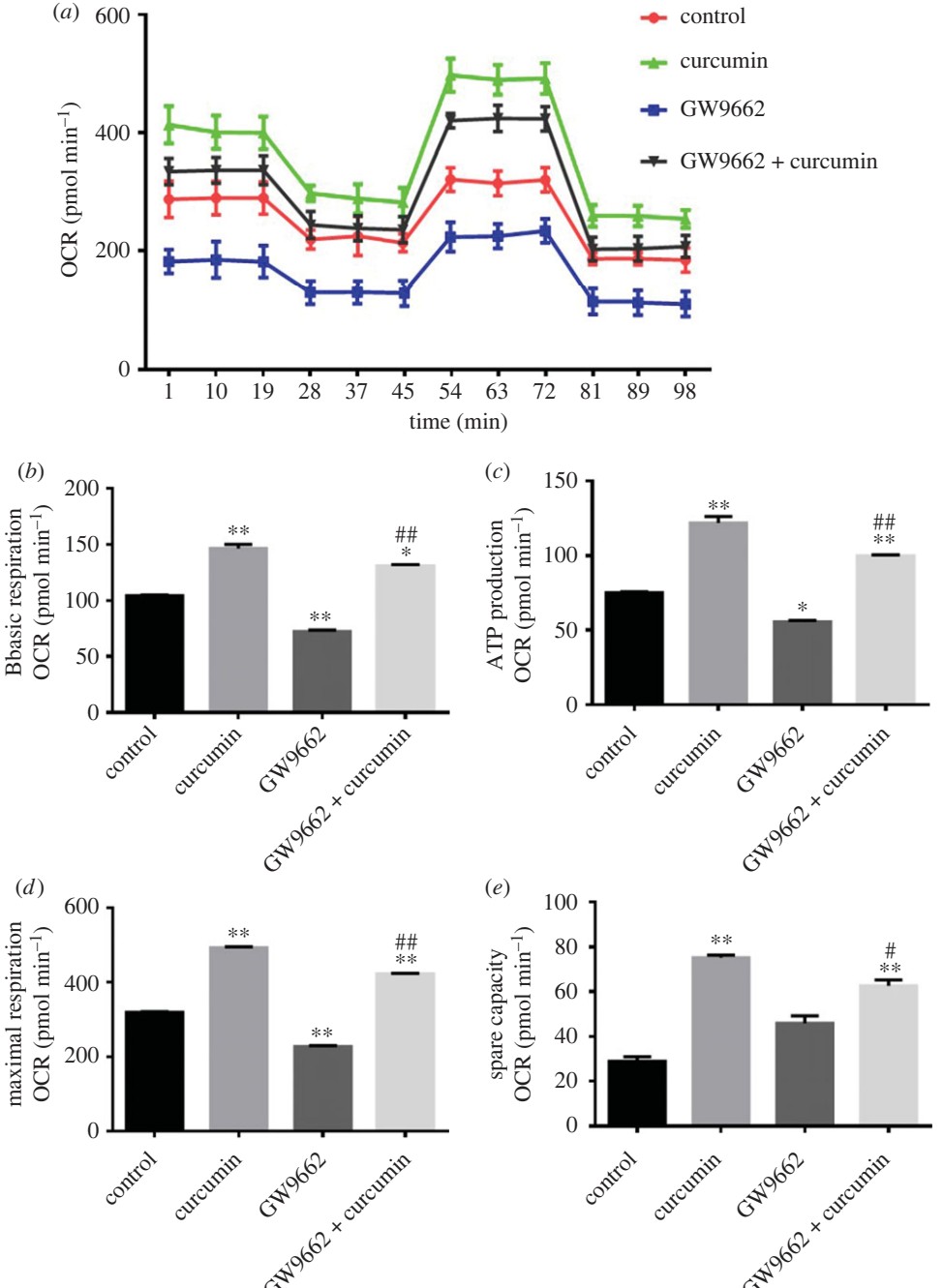

**Figure 4.** Curcumin altered OCR of mature adipocytes with or without a PPARγ antagonist. (*a*) Results of OCR curves in different groups. (*b*) Basal respiration. (*c*) ATP production. (*d*) Maximum respiration. (*e*) Spare capacity. Curcumin was used at 10 μM. Data are presented as the mean ± s.e.m. from three repeated experiments. $^*p < 0.05$, $^{**}p < 0.01$ compared with the untreated control, $^\#p < 0.05$, $^{\#\#}p < 0.01$ versus the curcumin-treated group.

compared to that in the control group (figure 3*f*). These results indicate that 10 μM curcumin significantly promotes mitochondrial respiratory function. By contrast, curcumin at doses of 20 and 35 μM inhibits mitochondrial respiratory function.

## 3.3. Curcumin stimulated mitochondrial respiration via activating PPARγ

Next, we examined the PPARγ alterations, a key factor in adipocyte differentiation induction [17], in response to curcumin treatment. As shown in figure 4, the PPARγ antagonist (GW9692) inhibited mitochondrial respiration in mature 3T3-L1 adipocytes. Furthermore, the PPARγ antagonist attenuated

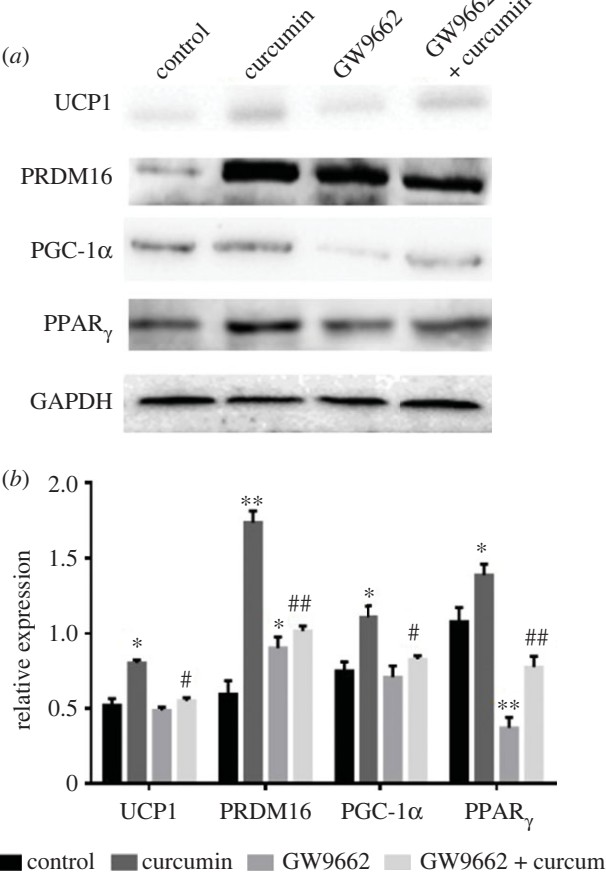

**Figure 5.** Effects of curcumin (10 µM) on the protein expression levels of UCP1, PRDM16, PGC-1α and PPARγ in 3T3-L1 differentiated adipocytes. UCP1, mitochondrial uncoupling protein 1 antibodies; PRDM16, PR domain protein 16; PGC-1α, peroxisome proliferator-activated receptor-γ coactivator-1α; PPARγ, peroxisome proliferator-activated receptor γ. Data are presented as the mean ± s.e.m. of $N = 3$ replicates. $^*p < 0.05$, $^{**}p < 0.01$ versus the untreated control, $^{\#}p < 0.05$, $^{\#\#}p < 0.01$ versus the curcumin-treated group.

the improvement effect of curcumin (10 µM) on mitochondrial respiration (figure 4a). Specifically, the PPARγ antagonist significantly reduced the basal mitochondrial respiration (figure 4b), ATP production (figure 4c), maximum respiration (figure 4d) and spare capacity (figure 4e) after supplementation with curcumin ($p < 0.01$). Taken together, these data indicate that curcumin significantly improves the mitochondrial respiratory capacity in part through PPARγ induction in mature adipocytes.

## 3.4. Effect of curcumin on white adipocytes browning

As demonstrated in figure 5, curcumin (10 µM) treatment increased the protein expression levels of UCP1, PRDM16, PGC-1α and PPARγ in 3T3-L1 differentiated adipocytes by 0.40-, 1.91-, 0.47- and 0.29-fold, respectively ($p < 0.05$). Furthermore, GW9662 (a PPARγ inhibitor) treatment attenuated the curcumin-induced upregulation of PPARγ activity to a level near that achieved with GW9662 alone. These results indicate that curcumin could partly enhance PPARγ RNA and protein expressions, and then exert further pharmacological functions. Additionally, GW9662 blocked the increased protein expressions of UCP1 and PRDM16 in response to curcumin treatment ($p < 0.05$), but it did not decrease the PGC-1α expression. These results show that curcumin induces the browning of white adipocytes through triggering PPARγ.

## 3.5. Effect of curcumin on white and brown adipose tissue in mice

As shown in figure 6, the WAT adipocytes were less organized (figure 6a) and the volume of fat cells was significantly larger in DIO group mice compared to those in normal group mice (figure 6c). After

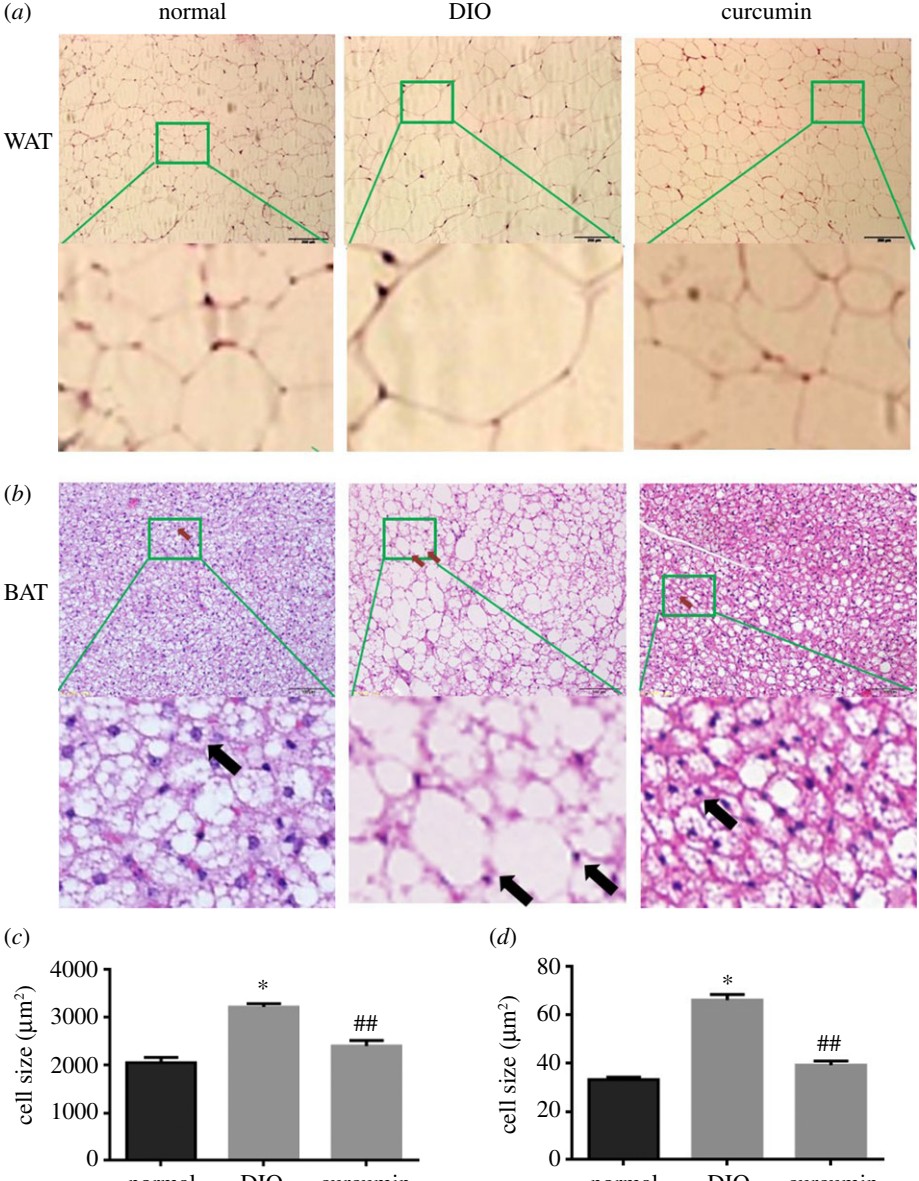

**Figure 6.** Effect of curcumin (50 mg kg$^{-1}$ d$^{-1}$) on adipose tissue in mice ($n = 6$). ($a,b$) Representative haematoxylin and eosin (H&E) staining of WAT and BAT to visualize lipid droplets and determine adipocyte size (200×). Scale bars signified 200 μM in ($a$) and 100 μM in ($c$). ($c,d$) The average adipocyte sizes in WAT and BAT were determined by counting 50 cells using ImageJ software. The green squares indicate the typical area while the arrows indicate the cell nuclei in adipose tissue. DIO, diet-induced obesity; WAT, white adipose tissue; BAT, brown adipose tissue. $^*p < 0.05$ versus the normal control group, $^{\#\#}p < 0.01$ versus the DIO group.

curcumin (50 mg kg$^{-1}$ d$^{-1}$) treatment, the cell sizes were markedly decreased. The nuclei (blue) and cytoplasm (pink) of adipose cells were squeezed to one side because of the emergence of lipid droplets in the DIO group compared with those in normal BAT (figure 6$b$). Moreover, the BAT architecture in obese mice demonstrated a higher number of lipid droplets. Curcumin treatment significantly decreased the fat cell size and attenuated lipid droplet accumulation compared with those in the DIO group (figure 6$d$). Interestingly, more visible cavity beige fat cells were observed in the curcumin treatment group than in the control group, which indicated that curcumin can inhibit the hypertrophy of white fat cells and promote the transition of white fat cells to brown fat cells.

## 3.6. Effect of curcumin on relevant mRNA expression in WAT and BAT

As shown in figure 7, the mRNA expression levels of UCP-1, PRDM16, PGC-1α and PPARγ in the WAT and BAT of the DIO group were notably lower than those in the normal group ($p < 0.05$). After curcumin (50 mg kg$^{-1}$ d$^{-1}$) treatment, the WAT mRNA expression levels of UCP1, PRDM16, PGC-1α and PPARγ

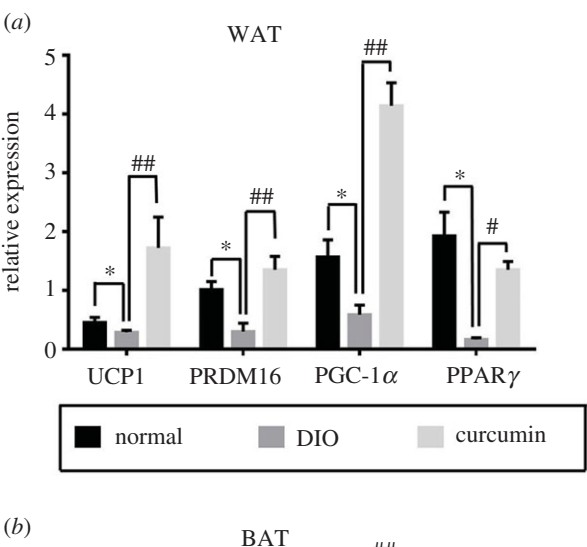

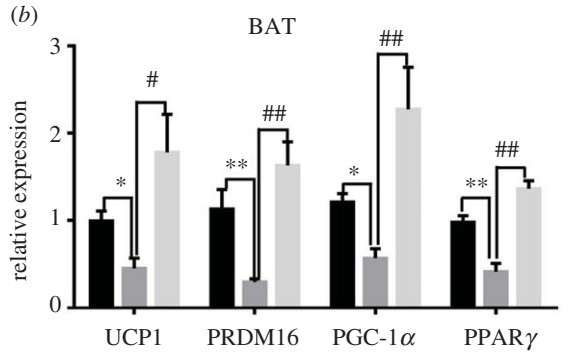

**Figure 7.** Effect of curcumin (50 mg kg$^{-1}$ d$^{-1}$) on relevant mRNA expression in white and brown adipose tissues ($n = 6$). Data are presented as the means $\pm$ s.e.m. in the normal, DIO and curcumin groups. $^{*}p < 0.05$, $^{**}p < 0.01$ versus the normal group, $^{\#}p < 0.05$, $^{\#\#}p < 0.01$ versus the DIO group.

were increased by 4.7-, 3.5-, 6.0- and 7.2-fold, respectively, compared with those in the DIO group ($p < 0.05$). The mRNA expression levels of UCP1, PRDM16, PGC-1α and PPARγ in the BAT were increased by 2.9-, 4.4-, 3.0- and 2.3-fold, respectively, compared with mice in the DIO group.

## 3.7. Effect of curcumin on relevant protein expression levels in WAT and BAT

As shown in figure 8, the WAT and BAT protein expression levels of PRDM16, PGC-1α and PPARγ were decreased in the DIO group compared with those in the normal group. However, the WAT protein expression levels of UCP1, PRDM16, PGC-1α and PPARγ in the curcumin treatment group were significantly increased compared with those in the DIO group ($p < 0.05$). Additionally, the expression levels of these proteins in BAT were higher in the curcumin treatment group than in the vehicle treated DIO group ($p < 0.05$).

# 4. Discussion

It is generally accepted that abdominal obese individuals exhibit a higher trend to develop metabolic diseases and cardiovascular complications [18]. The reason for this increase may lie in the negative correlation between body weight and insulin sensitivity [19]. Therefore, strategies to control obesity have direct applications in enhancing insulin sensitivity and deferring the onset or progression of type 2 diabetes. Recently, evidence from preclinical studies has demonstrated a diversity of pharmacological actions of curcumin, among which anti-inflammatory [20], antioxidant [9,21], anti-infectious [22] and cardioprotective effects [23] have attracted substantial scientific interest. It was demonstrated that curcumin could suppress inflammation through inhibiting the transcription factor NF-κB and regulating the expressions of pro-inflammatory genes, such as TNF, IL-1, IL-6, IL-8 and chemokines [24]. In addition, both animal and cellular models have shown that curcumin regulates

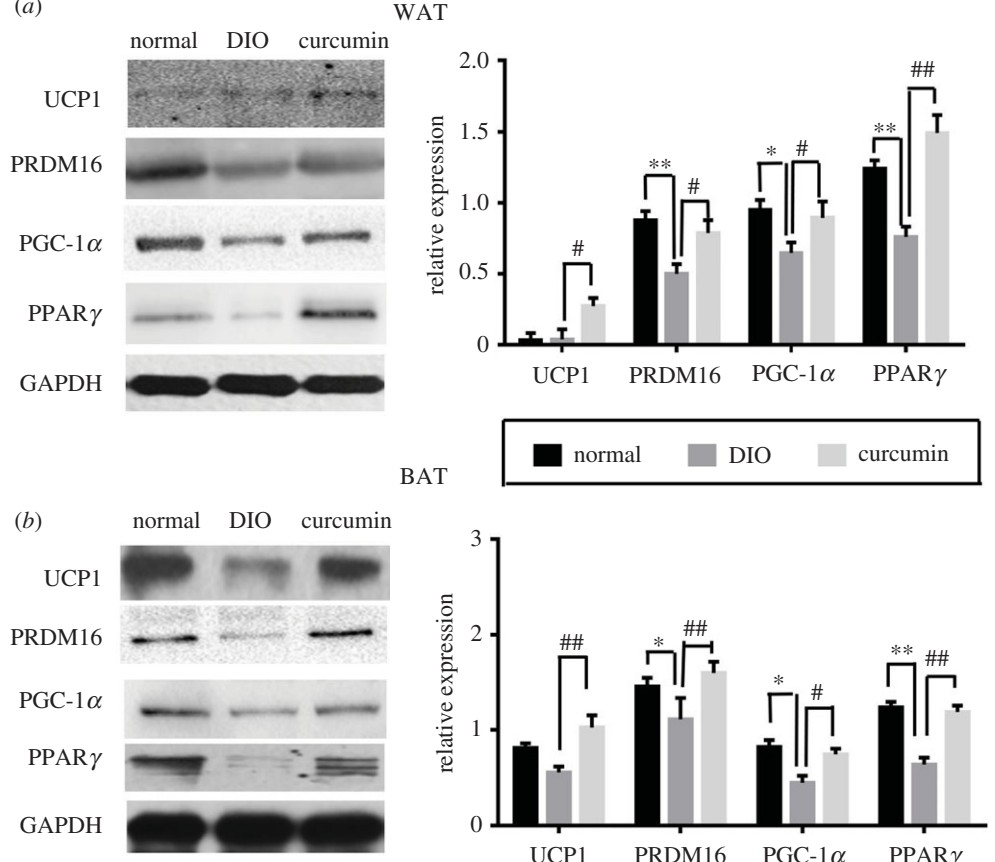

**Figure 8.** Effect of curcumin (50 mg kg$^{-1}$ d$^{-1}$) on relevant protein expression in white and brown adipose tissues ($n = 6$). Data are presented as the means ± s.e.m. in the normal, DIO and curcumin groups. $^*p < 0.05$, $^{**}p < 0.01$ versus the normal group, $^#p < 0.05$, $^{##}p < 0.01$ versus the DIO group.

the expressions of SOD and GSH, which helps to keep redox homeostasis [25]. Recently, extensive clinical trials have also revealed that curcumin has potential to protect against a wide variety of diseases, such as skin, respiratory, cardiovascular, gastrointestinal, nervous system and metabolic disorders [26]. Results from our previous research also illustrated that curcumin could relieve insulin resistance and decrease blood glucose levels in obese mice [13].

Adipose tissues are important insulin-sensitive peripheral tissues that are traditionally considered to primarily function as an energy store. When fuel is required, triglycerides in adipocytes are broken down into free fatty acids (FFAs), released into circulation, and further oxidized to provide energy for utilization [27]. Nevertheless, increasing evidence suggests that adipose tissues are also involved in other physiological processes, such as the secretion of TNFα and adiponectin [27,28]. Additionally, the function of brown adipocytes (minority adipocytes with irreplaceable roles) and metabolically beneficial effects of white adipocyte browning have attracted broad attention [28,29]. In the present study, the results of Oil Red O staining and triglyceride level in differentiated 3T3-L1 cells demonstrated that curcumin induced differentiation of the preadipocytes and stimulated the lipid droplets intracellular accumulation. Additionally, our *in vivo* results revealed that curcumin attenuated pathological damage and enhanced the browning process of adipose tissues in DIO mice, which corresponds with the results of previous studies [30].

Mitochondria are important organelles participating in the regulation of numerous cellular activities, including apoptosis, thermogenesis, reactive oxygen species production, redox and Ca$^{2+}$ homeostasis [31,32]. Mitochondrial function is vital to the functionality and viability of adipocytes. A recent concept emerged highlighting that mitochondrial dysfunction in adipocytes is tightly related to insulin resistance in obese and diabetic individuals [33,34]. And cellular oxygen consumption in cells is commonly acknowledged as a fundamental assessing indicator of mitochondrial function. Specifically, mitochondrial basic respiration includes coupled as well as the uncoupled mitochondrial oxygen consumption. As we know the former process produces ATP, and the latter process forms reactive

oxygen species, which is an agent involved in multiple physiological and pathological activities, when electrons leak out of respiratory chain to mitochondrial inner membrane and matrix. The maximal OCR is an indicator which represents the ability of mitochondria to reserve energy [35]. To examine the effect of curcumin on mitochondrial respiration, we first observed the metabolic effects of curcumin at different concentrations (10, 20 and 35 µM) on the mitochondrial respiratory status of 3T3-L1 adipocytes referring to the literature [30]. And we revealed that 10 µM curcumin significantly promoted mitochondrial respiration function and thereby improved adipocyte fuel utilization. However, higher doses of curcumin (20 and 35 µM) exerted adverse effects on mitochondrial respiratory function. It was reported that some natural products and compounds showed a non-monotonic dose response—bell-curves for stimulatory biological effects [36], which may be one reason for our findings. In addition, curcumin at a high concentration may affect the morphology of mitochondrial ridges and the permeability of the mitochondria membrane, thus inhibiting mitochondrial oxidative respiration. We will attempt to confirm the effect of curcumin on the morphology of mitochondria in the future.

PPARγ, a vital nuclear receptor, exerts a principal function in the regulating adipogenic differentiation and lipid metabolism [17,37]. So far, it has been proved that various genes, such as those encoding GLUT-2 and IRS-2, target PPARγ directly and are involved in extensive pathophysiological process, like glucose disposal [38]. Based on our previous findings [39,40], we further investigated whether the influence on mitochondrial dysfunction relied on PPARγ by incubating 3T3-L1 adipocytes with a PPARγ antagonist and/or curcumin. The PPARγ antagonist partially inhibited the ability of curcumin (10 µM) to improve mitochondrial respiration, which indicates that PPARγ may be the pivotal point for further exploring the mechanisms of curcumin in adipocytes although other mechanisms may also be involved.

PPARγ is a chief regulator that can promote the differentiation process of white adipocytes collaborating with multiple other regulators, such as PGC-1, CCAAT/enhancer binding proteinβ (C/EBPβ), phosphoenolpyruvate carboxykinase (PEPCK) and so on [41]. Accordingly, we further examined the molecular mechanisms by which curcumin exerts its effects, associating with PPARγ in adipocytes during the processes of white adipose browning and mitochondrial respiration. BAT specializes in the dissipation of energy as heat, which is dependent on UCP1, acting as a proton passageway in the inner mitochondrial membrane. The electrochemical proton gradient yield by protons leaking through this passageway is essential for ATP synthesis in mitochondria. Therefore, it has been assured that the increased UCP1 expression in WAT is a potential molecular strategy to conquer obesity [42]. During the browning process, beige or brite adipocytes in WAT switches to a brown fat cell-like phenotype, observed as UCP1+ adipocytes. PGC-1α is also an important nuclear transcription factor that controls the expression of UCP1 [43,44] and acts as a main regulator of mitochondrial biogenesis [45]. PRDM16, another significant molecule participating in the 'browning' process of WAT, is also rich in subcutaneous fat tissues. Coincidentally, mice in which PRDM16 was knocked down developed enlargement of subcutaneous fat tissues as well as decreased insulin sensitivity [46]. Moreover, PRDM16 activates UCP1 and PGC-1α through direct binding [47]. In our study, curcumin increased the protein expression levels of PPARγ, PGC-1α, UCP1 and PRDM16 in mature adipocytes. Similarly, curcumin upregulated the mRNA and protein expression levels of PPARγ, PGC-1α and UCP1 as well as the mRNA levels of PRDM16 in the WAT and BAT of high-fat DIO mice.

# 5. Conclusion

Taken together, the results of the present study demonstrated that curcumin may provide a new solution for controlling and managing obesity and other metabolic diseases. The underlying mechanism may be attributed to an improvement in mitochondrial function and promotion of adipogenesis via regulation of PPARγ expression, which further led to increased expression levels of the relevant transcription factors PGC-1α, UCP1 and PRDM16 in adipocytes. In the future, corresponding clinical trials are necessary to confirm these results.

Ethical. This study was carried out in compliance with the 'Guide for the Care and Use of Experimental Animals, Animal Care Committee of Beijing University of Chinese Medicine, China'. The protocol was approved by the Animal Care Committee of Beijing University of Chinese Medicine, China (no. BUCM-4-2016061701-3001).
Data accessibility. Data have been uploaded as electronic supplementary material, Excel file.
Authors' contributions. Z.D., P.Y. and Y.N. performed the experiments. Z.D. and P.Y. wrote the manuscript. G.S., Z.D. and G.J. conceived and designed the experiments. M.R., M.F., C.B. and J.Q. helped perform the animal experiments and

collected the samples. B.Y. and Z.J. participated in data analyses. B.Y. and L.J. contributed to the revision of the manuscript. All authors read and approved the final manuscript.

Competing interests. We declare we have no competing interests.

Funding. This study received the support from National Natural Science Foundation of China (grant nos. 81274041, 81503540 and 81273995), the Collaborative Innovation Project of Beijing Chaoyang district (grant no. CYXC1513).

Acknowledgements. We thank Qianqian Mu and Xin Fang for assistance with this study.

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
