## [Peer Review File · Royal Society Open Science]

Review History

RSOS-200974.R0 (Original submission)

Review form: Reviewer 1

Is the manuscript scientifically sound in its present form?

Yes

Are the interpretations and conclusions justified by the results?

Yes

Is the language acceptable?

Yes

Do you have any ethical concerns with this paper?

No

Have you any concerns about statistical analyses in this paper?

No

Recommendation?

Accept with minor revision (please list in comments)

Comments to the Author(s)

Mitochondrial dysfunction and adipocyte differentiation are thought to promote lipid accumulation in the development of obesity and diabetes. This manuscript reports the effects of curcumin on mitochondrial function and adipocyte browning and investigates potential molecular mechanisms using cultured cells and an obese mouse model. The authors demonstrate that curcumin at certain concentrations promotes the adipogenic differentiation of preadipocytes and mitochondrial oxygen consumption in mature adipocytes and suggest this is due to upregulation of UCP1, PRDM16, PPAR γ and PGC-1 α expression.

The manuscript would benefit from the following changes and additions:

Introduction: In the introduction, the authors describe how curcumin is reported to prevent and treat a wide range of diseases, however they don't state whether this evidence has come from preclinical studies or human trials; consequently the statement is currently misleading. The references cited appear to be specific preclinical studies on the different disease areas rather than reviews, which would have more weighting. This statement needs more context and the authors should describe what is known about the effects of curcumin in clinical trials, particularly trials that might be relevant to the management of obesity.

Curcumin has notoriously low bioavailability due to poor absorption from the gastrointestinal tract and rapid metabolism. Consequently, even when high doses are given to humans and animals, only low systemic concentrations of curcumin are achieved. The authors need to acknowledge this issue and explain how the concentrations and doses they have used in their in vitro and in vivo studies translate to humans - particularly the concentration range of 10-35 μ M - is this attainable in humans?

Methods, Cell counting section 2.3: What was the curcumin dissolved in and what was the concentration of the initial stock solution? This is important because the authors mention in the discussion that curcumin may have precipitated out in cell culture at the higher concentrations.

Section 2.6 Animals: Details of the methods need clarifying. It seems like the obese mice on the high fat diet for 12 weeks were then used in the curcumin study for an additional 8 weeks. If this was the case, please state how many mice were put on high fat diet at the start - it says n=10 for the normal diet but there is no number given for the high fat group - presumably this was at least 20 if 10 mice per group were needed for the curcumin part of the study? When the mice started receiving curcumin and saline were they maintained on the high fat diet? Please include this information.

Also, were the mice on normal fat diet for 12 weeks then kept on study for a further 8 weeks so they could be directly compared with the obese mice? This isn't mentioned.

Please also justify the choice of curcumin dose given to the animals - how does this translate to a human equivalent dose?

Please provide details in the methods on how many mice were used for each type of analysis (histological, gene expression, protein expression), whether different animals were selected for each type of analysis and if so why? Also, how were the animals selected - was it random?

Results, Section 3.1: The authors should explain why they were investigating the effect of curcumin on viability of preadipocytes - what is the significance of the reduced viability observed at concentrations of 50 and 75 μ M?

Section 3.1: The authors state that ‘Microscopic observation revealed that curcumin accelerated adipocyte differentiation and enhanced the amount of lipid droplets in the cells.’ Could they please explain in the text how they know that the effect is on differentiation specifically, rather than curcumin having a biochemical effect on the production of lipids, or other parameter that may influence the lipid droplets in cells. Is there something about the morphology of the cells that could be highlighted in the figure?

Section 3.3, last sentence: Based on the results with the PPAR γ antagonist the authors state that ‘10 μ M curcumin significantly improves the mitochondrial respiratory capacity of mature adipocytes through PPAR γ induction’. Whilst I would agree that the findings support the idea that curcumin induces PPAR γ , the antagonist has a relatively small effect when used in combination with curcumin and it does not completely cancel out the increases seen with curcumin relative to the control incubations. Therefore, this statement may be too strong – the effects may be partly through PPAR γ but other mechanisms could also be involved and this should be acknowledged.

Section 3.4: The statement ‘GW9662 (a PPAR γ inhibitor) treatment attenuated the curcumin-induced upregulation of PPAR γ activity..’ needs explaining. How are the protein expression levels shown in Figure 5 providing a measure of PPAR γ activity?

Section 3.4, last sentence: The effect of curcumin on browning-specific genes was not assessed in this experiment – please correct.

Section 3.5: The authors mention how ‘the nuclei and cytoplasm of adipose cells were squeezed to one side’ and that there was a ‘higher number of lipid droplets’ – it would be really helpful if these features could be clearly indicated on the images in Figure 6 to illustrate the effects described. Similarly, please highlight the beige fat cells in the curcumin treated group – it is not currently clear where these cells are in the images shown.

Discussion, first paragraph: As indicated for the introduction, please also give more context in the discussion when describing the pharmacological actions of curcumin – it needs to be clear where the authors are referring to preclinical effects versus established effects in humans in vivo.

Discussion, 2nd paragraph: please explain what is meant by my ‘minority adipocytes with irreplaceable roles’.

Also, in the discussion please elaborate on the evidence that supports the fact curcumin induced differentiation of adipocytes.

In the discussion, the authors suggest that precipitation of curcumin at high concentrations may account for the adverse effects on mitochondrial respiratory function. This is a concern, as it raises questions about the validity of the data obtained at concentrations exceeding 10 μ M. In the methods and also here in the discussion the authors need to explain what measures they took to try and avoid precipitation of curcumin and also what evidence they have to support the suggestion – was there a visible precipitate for example? It would also be worth discussing how their chosen concentration range compares with the literature on in vitro curcumin experiments. Another possibility worth considering is a non-linear dose response; bell-shaped or U-shaped dose responses have been reported for various biological activities of other naturally occurring compounds – could this be the case for curcumin here? This should also be discussed.

Discussion, page 13, line 55/56: The statement ‘directly or potential for further pathophysiological process’ doesn’t make sense. Please revise.

Page 14, 1st line: please acknowledge that the PPAR γ antagonist did not completely inhibit the effects of curcumin on mitochondrial respiration, so other mechanisms may also be involved.

Figure 5 legend: delete 'DIO, diet-induced obesity'. This figure contains in vitro data so this abbreviation is not necessary.

Review form: Reviewer 2

Is the manuscript scientifically sound in its present form?

No

Are the interpretations and conclusions justified by the results?

No

Is the language acceptable?

Yes

Do you have any ethical concerns with this paper?

No

Have you any concerns about statistical analyses in this paper?

Yes

Recommendation?

Major revision is needed (please make suggestions in comments)

Comments to the Author(s)

This study describes the effect of curcumin on adipocyte metabolism in vitro and in vivo. Curcumin promoted adipogenesis in 3T3-L1 cells and affected mitochondrial respiration (low and high dose increased and decreased respiration, respectively). Low dose of curcumin increased expression of Ucp1 and genes related to adipogenesis. In vivo, curcumin decreased lipid accumulation in both WAT and BAT and increased expression of Ucp1 and genes related to adipogenesis.

Although the in vitro study was done properly, interpretation of the results concerning the implication of PPAR γ is misleading. I agree that the PPAR γ antagonist attenuated the effect of curcumin (figure 4), but most of the effects was still present. It suggests that most of the curcumin effects on respiration are PPAR γ -independent. Page 10 line 43, the interpretation should be that curcumin improves mitochondrial respiratory capacity of adipocyte in part through PPAR γ induction. The authors could measure more PPAR γ target gene expression (for instance, Angptl4, Cidea, Plin1) to make a stronger case on PPAR γ involvement.

I am concern that the dosage of curcumin could be toxic to the mice, which would explain the increase in lipolysis in WAT (in the previous publication) and increase FFA-induced browning in WAT. The authors should perform food intake experiments to demonstrate that the curcumin concentration does not have a detrimental effect on food consumption.

In this line, how the authors reconcile that the in vitro study showed that curcumin increase lipid accumulation while decreasing adipocyte size in the in vivo study?

Minor points

Page 14 line 12: it is not clear what "active factors" means.

Page 14 line 19 and later: same for "Passageway"

Decision letter (RSOS-200974.R0)

Dear Dr Zhao

The Editors assigned to your paper RSOS-200974 "Curcumin improves adipocytes browning and mitochondrial function in 3T3-L1 cells and obese rodent model" have now received comments from reviewers and would like you to revise the paper in accordance with the reviewer comments and any comments from the Editors. Please note this decision does not guarantee eventual acceptance.

Please submit your revised manuscript and required files (see below) no later than 21 days from today's (ie 06-Oct-2020) date. Note: the ScholarOne system will 'lock' if submission of the revision is attempted 21 or more days after the deadline. If you do not think you will be able to meet this deadline please contact the editorial office immediately.

on behalf of Prof Catrin Pritchard (Subject Editor)
openscience@royalsociety.org

Associate Editor Comments to Author (Dr Catrin Pritchard):
Associate Editor: 1
Comments to the Author:

The reviewers both agree the manuscript is potentially interesting but suggest a number of revisions should be made

Reviewer comments to Author:

Reviewer: 1

Comments to the Author(s)

Mitochondrial dysfunction and adipocyte differentiation are thought to promote lipid accumulation in the development of obesity and diabetes. This manuscript reports the effects of curcumin on mitochondrial function and adipocyte browning and investigates potential molecular mechanisms using cultured cells and an obese mouse model. The authors demonstrate that curcumin at certain concentrations promotes the adipogenic differentiation of preadipocytes and mitochondrial oxygen consumption in mature adipocytes and suggest this is due to upregulation of UCP1, PRDM16, PPAR γ and PGC-1 α expression.

The manuscript would benefit from the following changes and additions:

Introduction: In the introduction, the authors describe how curcumin is reported to prevent and treat a wide range of diseases, however they don't state whether this evidence has come from preclinical studies or human trials; consequently the statement is currently misleading. The references cited appear to be specific preclinical studies on the different disease areas rather than reviews, which would have more weighting. This statement needs more context and the authors should describe what is known about the effects of curcumin in clinical trials, particularly trials that might be relevant to the management of obesity.

Curcumin has notoriously low bioavailability due to poor absorption from the gastrointestinal tract and rapid metabolism. Consequently, even when high doses are given to humans and animals, only low systemic concentrations of curcumin are achieved. The authors need to acknowledge this issue and explain how the concentrations and doses they have used in their in vitro and in vivo studies translate to humans - particularly the concentration range of 10-35 μ M - is this attainable in humans?

Methods, Cell counting section 2.3: What was the curcumin dissolved in and what was the concentration of the initial stock solution? This is important because the authors mention in the discussion that curcumin may have precipitated out in cell culture at the higher concentrations.

Section 2.6 Animals: Details of the methods need clarifying. It seems like the obese mice on the high fat diet for 12 weeks were then used in the curcumin study for an additional 8 weeks. If this was the case, please state how many mice were put on high fat diet at the start - it says n=10 for the normal diet but there is no number given for the high fat group - presumably this was at least 20 if 10 mice per group were needed for the curcumin part of the study? When the mice started receiving curcumin and saline were they maintained on the high fat diet? Please include this information.

Also, were the mice on normal fat diet for 12 weeks then kept on study for a further 8 weeks so they could be directly compared with the obese mice? This isn't mentioned.

Please also justify the choice of curcumin dose given to the animals - how does this translate to a human equivalent dose?

Please provide details in the methods on how many mice were used for each type of analysis (histological, gene expression, protein expression), whether different animals were selected for each type of analysis and if so why? Also, how were the animals selected - was it random?

Results, Section 3.1: The authors should explain why they were investigating the effect of curcumin on viability of preadipocytes – what is the significance of the reduced viability observed at concentrations of 50 and 75 μM ?

Section 3.1: The authors state that ‘Microscopic observation revealed that curcumin accelerated adipocyte differentiation and enhanced the amount of lipid droplets in the cells.’ Could they please explain in the text how they know that the effect is on differentiation specifically, rather than curcumin having a biochemical effect on the production of lipids, or other parameter that may influence the lipid droplets in cells. Is there something about the morphology of the cells that could be highlighted in the figure?

Section 3.3, last sentence: Based on the results with the PPAR γ antagonist the authors state that ‘10 μM curcumin significantly improves the mitochondrial respiratory capacity of mature adipocytes through PPAR γ induction’. Whilst I would agree that the findings support the idea that curcumin induces PPAR γ , the antagonist has a relatively small effect when used in combination with curcumin and it does not completely cancel out the increases seen with curcumin relative to the control incubations. Therefore, this statement may be too strong – the effects may be partly through PPAR γ but other mechanisms could also be involved and this should be acknowledged.

Section 3.4: The statement ‘GW9662 (a PPAR γ inhibitor) treatment attenuated the curcumin-induced upregulation of PPAR γ activity..’ needs explaining. How are the protein expression levels shown in Figure 5 providing a measure of PPAR γ activity?

Section 3.4, last sentence: The effect of curcumin on browning-specific genes was not assessed in this experiment – please correct.

Section 3.5: The authors mention how ‘the nuclei and cytoplasm of adipose cells were squeezed to one side’ and that there was a ‘higher number of lipid droplets’ – it would be really helpful if these features could be clearly indicated on the images in Figure 6 to illustrate the effects described. Similarly, please highlight the beige fat cells in the curcumin treated group – it is not currently clear where these cells are in the images shown.

Discussion, first paragraph: As indicated for the introduction, please also give more context in the discussion when describing the pharmacological actions of curcumin – it needs to be clear where the authors are referring to preclinical effects versus established effects in humans *in vivo*.

Discussion, 2nd paragraph: please explain what is meant by ‘minority adipocytes with irreplaceable roles’.

Also, in the discussion please elaborate on the evidence that supports the fact curcumin induced differentiation of adipocytes.

In the discussion, the authors suggest that precipitation of curcumin at high concentrations may account for the adverse effects on mitochondrial respiratory function. This is a concern, as it raises questions about the validity of the data obtained at concentrations exceeding 10 μM . In the methods and also here in the discussion the authors need to explain what measures they took to try and avoid precipitation of curcumin and also what evidence they have to support the suggestion – was there a visible precipitate for example? It would also be worth discussing how their chosen concentration range compares with the literature on *in vitro* curcumin experiments. Another possibility worth considering is a non-linear dose response; bell-shaped or U-shaped dose responses have been reported for various biological activities of other naturally occurring compounds – could this be the case for curcumin here? This should also be discussed.

Discussion, page 13, line 55/56: The statement 'directly or potential for further pathophysiological process' doesn't make sense. Please revise.

Page 14, 1st line: please acknowledge that the PPAR γ antagonist did not completely inhibit the effects of curcumin on mitochondrial respiration, so other mechanisms may also be involved.

Figure 5 legend: delete 'DIO, diet-induced obesity'. This figure contains in vitro data so this abbreviation is not necessary.

Reviewer: 2

Comments to the Author(s)

This study describes the effect of curcumin on adipocyte metabolism in vitro and in vivo. Curcumin promoted adipogenesis in 3T3-L1 cells and affected mitochondrial respiration (low and high dose increased and decreased respiration, respectively). Low dose of curcumin increased expression of Ucp1 and genes related to adipogenesis. In vivo, curcumin decreased lipid accumulation in both WAT and BAT and increased expression of Ucp1 and genes related to adipogenesis.

Although the in vitro study was done properly, interpretation of the results concerning the implication of PPAR γ is misleading. I agree that the PPAR γ antagonist attenuated the effect of curcumin (figure 4), but most of the effects was still present. It suggests that most of the curcumin effects on respiration are PPAR γ -independent. Page 10 line 43, the interpretation should be that curcumin improves mitochondrial respiratory capacity of adipocyte in part through PPAR γ induction. The authors could measure more PPAR γ target gene expression (for instance, Angptl4, Cidea, Plin1) to make a stronger case on PPAR γ involvement.

I am concern that the dosage of curcumin could be toxic to the mice, which would explain the increase in lipolysis in WAT (in the previous publication) and increase FFA-induced browning in WAT. The authors should perform food intake experiments to demonstrate that the curcumin concentration does not have a detrimental effect on food consumption.

In this line, how the authors reconcile that the in vitro study showed that curcumin increase lipid accumulation while decreasing adipocyte size in the in vivo study?

Minor points

Page 14 line 12: it is not clear what "active factors" means.

Page 14 line 19 and later: same for "Passageway"

===PREPARING YOUR MANUSCRIPT===

Please ensure that you include an acknowledgements' section before your reference list/bibliography. This should acknowledge anyone who assisted with your work, but does not

qualify as an author per the guidelines at <https://royalsociety.org/journals/ethics-policies/openness/>.

===PREPARING YOUR REVISION IN SCHOLARONE===

- Ensure that your data access statement meets the requirements at <https://royalsociety.org/journals/authors/author-guidelines/#data>. You should ensure that you cite the dataset in your reference list. If you have deposited data etc in the Dryad repository, please include both the 'For publication' link and 'For review' link at this stage.
- If you are requesting an article processing charge waiver, you must select the relevant waiver option (if requesting a discretionary waiver, the form should have been uploaded at Step 3 'File upload' above).
- If you have uploaded ESM files, please ensure you follow the guidance at <https://royalsociety.org/journals/authors/author-guidelines/#supplementary-material> to include a suitable title and informative caption. An example of appropriate titling and captioning may be found at https://figshare.com/articles/Table_S2_from_Is_there_a_trade-off_between_peak_performance_and_performance_breadth_across_temperatures_for_aerobic_scope_in_teleost_fishes_/3843624.

Author's Response to Decision Letter for (RSOS-200974.R0)

See Appendix A.

RSOS-200974.R1 (Revision)

Review form: Reviewer 2

Is the manuscript scientifically sound in its present form?

Yes

Are the interpretations and conclusions justified by the results?

Yes

Is the language acceptable?

Yes

Do you have any ethical concerns with this paper?

No

Have you any concerns about statistical analyses in this paper?

No

Recommendation?

Accept as is

Comments to the Author(s)

The authors have answered most of my critics. I don't have further comment.

Decision letter (RSOS-200974.R1)

Dear Dr Zhao,

It is a pleasure to accept your manuscript entitled "Curcumin improves adipocytes browning and mitochondrial function in 3T3-L1 cells and obese rodent model" in its current form for publication in Royal Society Open Science. The comments of the reviewers who reviewed your manuscript are included at the foot of this letter.

on behalf of Dr Catrin Pritchard (Subject Editor)
openscience@royalsociety.org

Reviewer comments to Author:

Reviewer: 2
Comments to the Author(s)
The authors have answered most of my critics. I don't have further comment.

Appendix A

30-November-2020

Ref. RSOS-200974

Title: Curcumin improves mitochondrial function and browning of adipocytes in 3T3-L1 cells and high-fat diet-induced obese mice

Dear editors and reviewers,

Thank you very much for giving us an opportunity to revise our manuscript entitled “Curcumin improves mitochondrial function and browning of adipocytes in 3T3-L1 cells and high-fat diet-induced obese mice” (ID: RSOS-200974). We would like to express our sincere thanks for the constructive comments and suggestions to our submission, which greatly help us to improve the quality of our manuscript.

We have read the referees’ comments very carefully and revised our manuscript according to the comments and RSOS submission style. We hope, with these improvements, our manuscript will be accepted for publication.

Please kindly checked the **revised MS with highlighted text**, as well as a **point-by-point response** to the reviewers’ comments.

If there any other questions, please don’t be hesitate to contact us.

Best wishes,

Sihua Gao, MD

Traditional Chinese Medicine School

Beijing University of Chinese Medicine

Beijing 100029, PR China

Phone: (8610) 6428-6929

Fax: (8610) 6428-6929

Email: gaosh@bucm.edu.cn

Point-by-point Response to Reviewers

Reviewer: 1

Firstly, we really appreciated for you helpful comments. The sentence with underline is the original text from the revised manuscript.

1. Introduction: In the introduction, the authors describe how curcumin is reported to prevent and treat a wide range of diseases, however they don't state whether this evidence has come from preclinical studies or human trials; consequently, the statement is currently misleading. The references cited appear to be specific preclinical studies on the different disease areas rather than reviews, which would have more weighting. This statement needs more context and the authors should describe what is known about the effects of curcumin in clinical trials, particularly trials that might be relevant to the management of obesity.

Response:

Thank you very much for your helpful suggestion. We have revised this part as follows:

“Accumulating evidence demonstrated that could prevent and treat a wide range of diseases by its anti-oxidant and anti-inflammatory properties in preclinical trials, for instance arthritis, cancer, neurodegenerative disorders, metabolic syndrome, cardiovascular diseases, diabetes and so on [8, 9]. and the same potential has been validated by many clinical trials [10]. Specifically, curcumin could regulate various molecular targets, such as nuclear factor κ B (NF- κ B), tumor necrosis factor - α (TNF α), monocyte chemoattractant protein - 1 (MCP - 1), which might contribute to its therapeutic potential in the management of obesity and diabetes [11]. Recently, curcumin was demonstrated to regulate inflammation and restore redox homeostasis among postpubescent overweight and obese girl adolescents [12].”

8. Hewlings S J, Kalman D S. 2017 Curcumin: A Review of its Effects on Human Health. Foods. 6. doi:10.3390/foods6100092

-
9. Sevastre-Berghian A C, Fagarasan V, Toma V A, Baldea I, Olteanu D, Moldovan R, et al. 2017 Curcumin reverses the Diazepam-Induced cognitive impairment by modulation of oxidative stress and ERK 1/2/NF-kappaB pathway in brain. Oxid Med Cell Longev. 2017, 3037876. doi:10.1155/2017/3037876
 10. Olotu F, Agoni C, Soremekun O, Soliman M. 2020 An Update on the Pharmacological Usage of Curcumin: Has it Failed in the Drug Discovery Pipeline? Cell Biochem Biophys. 78, 267-289. doi:10.1007/s12013-020-00922-5
 11. Bradford P G. 2013 Curcumin and obesity. Biofactors. 39, 78-87. doi:10.1002/biof.1074
 12. Saraf-Bank S, Ahmadi A, Paknahad Z, Maracy M, Nourian M. 2019 Effects of curcumin supplementation on markers of inflammation and oxidative stress among healthy overweight and obese girl adolescents: A randomized placebo-controlled clinical trial. Phytother Res. 33, 2015-2022. doi:10.1002/ptr.6370

2. *Curcumin has notoriously low bioavailability due to poor absorption from the gastrointestinal tract and rapid metabolism. Consequently, even when high doses are given to humans and animals, only low systemic concentrations of curcumin are achieved. The authors need to acknowledge this issue and explain how the concentrations and doses they have used in their in vitro and in vivo studies translate to humans – particularly the concentration range of 10-35 μ M – is this attainable in humans?*

Response:

Thank. We totally agree with the helpful suggestions.

Many preclinical and clinical studies showed curcumin exhibits low systemic bioavailability following oral dosing. However, it has also been proved that the pharmacologically active concentration of curcumin could be achieved in colorectal tissue in patients taking curcumin orally [1]. In addition, the results from clinical trials confirmed that a very low amount of circulating curcumin at steady-state may exhibit remarkable therapeutic effect although it has notoriously low bioavailability due to poor absorption and rapid metabolism [2].

Based on the clinical experience and literature, we have set the in vivo dosage of curcumin as 50 mg kg⁻¹ d⁻¹ for mice, which approximately equals to 240 mg per

day when translated to humans [3]. The administration of crude curcumin displayed a wide-ranging serum concentration from 1 to 3200 ng/mL depending on given dose (2 to 10g) and subject's physiology [4]. However, how serum concentration of curcumin compares to what is seen in cell culture conditions are still to be explored. In summary, the concentration of curcumin used in the in vitro experiments of the present study, 10-35 μ M, was determined based on our previous study and literature [5]. The revision was made and shown in the following response to comment 5 & 15.

[1] Sharma, Ricky A., William P. Steward, and Andreas J. Gescher. "Pharmacokinetics and pharmacodynamics of curcumin." *The molecular targets and therapeutic uses of curcumin in health and disease*. Springer, Boston, MA, 2007. 453-470

[2] Kunnumakkara, Ajaikumar B., et al. "Is curcumin bioavailability a problem in humans: lessons from clinical trials." *Expert opinion on drug metabolism & toxicology* 15.9 (2019): 705-733

[3] Nair, Anroop B., and Shery Jacob. "A simple practice guide for dose conversion between animals and human." *Journal of basic and clinical pharmacy* 7.2 (2016): 27

[4] Dei Cas, Michele, and Riccardo Ghidoni. "Dietary curcumin: Correlation between bioavailability and health potential." *Nutrients* 11.9 (2019): 2147

[5] Lone, J., et al. "Science Direct Curcumin induces brown fat-like phenotype in 3T3-L1 and primary white adipocytes." *J. Nutr. Biochem* 1.10 (2015).

3. *Methods, Cell counting section 2.3: What was the curcumin dissolved in and what was the concentration of the initial stock solution? This is important because the authors mention in the discussion that curcumin may have precipitated out in cell culture at the higher concentrations.*

Response:

We sincerely appreciate your comments. We have added the following content in the methods section:

“Curcumin was dissolved in Dimethyl sulfoxide (DMSO) and prepared as a 10 mM stock solution. Then the stock was diluted in the medium to the indicated concentration for the experiment.”

As for the mentioned precipitation in the discussion section, we are really sorry for the misleading. Precipitation of curcumin is only a possible reason to explain the adverse effects on mitochondrial respiratory function. In fact, we did not observe the precipitate in curcumin solution at high concentrations under the microscope. We only guessed there might be some invisible precipitation. To avoid misunderstanding, we have deleted this sentence in the revision.

4. Section 2.6 Animals: Details of the methods need clarifying. It seems like the obese mice on the high fat diet for 12 weeks were then used in the curcumin study for an additional 8 weeks. If this was the case, please state how many mice were put on high fat diet at the start – it says n=10 for the normal diet but there is no number given for the high fat group – presumably this was at least 20 if 10 mice per group were needed for the curcumin part of the study? When the mice started receiving curcumin and saline were they maintained on the high fat diet? Please include this information. Also, were the mice on normal fat diet for 12 weeks then kept on study for a further 8 weeks so they could be directly compared with the obese mice? This isn't mentioned.

Response:

We sincerely appreciate your comments. Initially, 26 mice were exposed on the high fat diet while 10 was on the normal diet. After 12 weeks, the 20 obese mice were randomly divided into two groups, with 10 mice in each. During the treatment, all mice maintained on the original diet, normal mice on the standard chow diet while obese mice on the high fat diet. In addition, the mice in the normal control group kept on for another 8 weeks to make further comparison with the obese mice. These were clarified in our manuscript in the animal section.

5. *Please also justify the choice of curcumin dose given to the animals – how does this translate to a human equivalent dose?*

Response:

Thank you very much for your comments. Based on clinical experience and references, we have set the in vivo dosage of curcumin as 50 mg kg⁻¹ d⁻¹ for mice, which approximately equals 240 mg per day when translated to humans (60kg) based on the body surface area [1].

[1] Nair, Anroop B., and Shery Jacob. "A simple practice guide for dose conversion between animals and human." *Journal of basic and clinical pharmacy* 7.2 (2016): 27

6. *Please provide details in the methods on how many mice were used for each type of analysis (histological, gene expression, protein expression), whether different animals were selected for each type of analysis and if so why? Also, how were the animals selected – was it random?*

Response:

Thank you very much for your comments. We also added the statement in the animal section as follows:

“Six samples were selected randomly from each group for further analysis of histological evaluation, and gene and protein expressions. And for each type of molecular biology analysis, the adipose tissues were from the same mice.”

7. *Results, Section 3.1: The authors should explain why they were investigating the effect of curcumin on viability of preadipocytes – what is the significance of the reduced viability observed at concentrations of 50 and 75 μM?*

Response:

We sincerely appreciate your comments. We wanted to determine a proper range of curcumin concentration under the condition that curcumin did not show cytotoxicity in preadipocytes, and the dosages will be used for the further experiments. The reduction of cell viability with curcumin concentration of 50 and 75 μ M may indicate its cytotoxicity.

8. *Section 3.1: The authors state that 'Microscopic observation revealed that curcumin accelerated adipocyte differentiation and enhanced the amount of lipid droplets in the cells.' Could they please explain in the text how they know that the effect is on differentiation specifically, rather than curcumin having a biochemical effect on the production of lipids, or other parameter that may influence the lipid droplets in cells. Is there something about the morphology of the cells that could be highlighted in the figure?*

Response:

We conducted 3T3-L1 differentiation experiment with a cocktail method in the presence and absence of curcumin. And Oil Red O (ORO) staining was performed to evaluate the extent of adipocyte differentiation. We revised our manuscript as follows:

“The results of Oil Red O staining revealed that different concentrations of curcumin accelerated adipocyte differentiation and increased the numbers of mature adipocytes (Figure 2B). In addition, curcumin could increase the triglyceride level in differentiated 3T3-L1 cells (Figure 2C). These results suggest that curcumin could influence lipid profiles through promotion of adipocyte differentiation.” We highlighted some specific differentiated adipocytes (arrows) in figure 2B in the revised manuscript.

Moreover, we found that PPAR γ , a factor is essential for adipogenesis, was affected by curcumin. Collectively, we inferred that an increase in lipid production in response to curcumin treatment was partly due to its effect on adipocytes differentiation.

9. Section 3.3, last sentence: *Based on the results with the PPAR γ antagonist the authors state that '10 μ M curcumin significantly improves the mitochondrial respiratory capacity of mature adipocytes through PPAR γ induction'. Whilst I would agree that the findings support the idea that curcumin induces PPAR γ , the antagonist has a relatively small effect when used in combination with curcumin and it does not completely cancel out the increases seen with curcumin relative to the control incubations. Therefore, this statement may be too strong – the effects may be partly through PPAR γ but other mechanisms could also be involved and this should be acknowledged.*

Response:

Thank you very much for your valuable suggestion. We have revised our statement as follows:

“these data indicate that curcumin significantly improves the mitochondrial respiratory capacity in part through PPAR γ induction in mature adipocytes”

10. Section 3.4: The statement ‘GW9662 (a PPAR γ inhibitor) treatment attenuated the curcumin-induced upregulation of PPAR γ activity.’ needs explaining. How are the protein expression levels shown in Figure 5 providing a measure of PPAR γ activity?

Response:

Thank you very much for your suggestion. Sorry for the misunderstanding of the activity. We have revised the description as follows:

“It indicated that curcumin could partly enhance PPAR γ mRNA and protein expressions, and then exert further pharmacological functions.”

11. Section 3.4, last sentence: The effect of curcumin on browning-specific genes was not assessed in this experiment – please correct.

Response:

Thank you very much for your careful inspection. We have revised this part as follows:

“.....curcumin induces the browning of white adipocytes through triggering PPAR γ .”

12. Section 3.5: The authors mention how ‘the nuclei and cytoplasm of adipose cells were squeezed to one side’ and that there was a ‘higher number of lipid droplets’ – it would be really helpful if these features could be clearly indicated on the images in Figure 6 to illustrate the effects described. Similarly, please highlight the beige fat cells in the curcumin treated group – it is not currently clear where these cells are in the images shown.

Response:

We would like to thank the reviewer's insightful comment. The whole picture of figure 6B that the nuclei and cytoplasm of adipose cells were squeezed to one side and there was a higher number of lipid droplets in the BAT of DIO group compared to Normal group. And as you suggested, we have revised Figure 6A & B by highlighting the key parts.

13. Discussion, first paragraph: As indicated for the introduction, please also give more context in the discussion when describing the pharmacological actions of curcumin – it needs to be clear where the authors are referring to preclinical effects versus established effects in humans *in vivo*.

Response:

Thank you very much for your suggestion. We have revised this by adding the content below:

“It was demonstrated that curcumin could suppress inflammation through inhibiting the transcription factor NF- κ B and regulating the expressions of pro-

inflammatory gene, such as TNF, IL-1, IL-6, IL-8, and chemokines [24]. In addition, both animal and cellular models have shown that curcumin regulates the expression of SOD and GSH, which helps to keep redox homeostasis [25]. Recently, extensive clinical trials have also revealed that curcumin has potential to protect against a wide variety of diseases, such as skin, respiratory, cardiovascular, gastrointestinal, nervous system and metabolic disorders [26].”

24. Menon V P, Sudheer A R. 2007 Antioxidant and anti-inflammatory properties of curcumin. Adv Exp Med Biol. 595, 105-125. doi:10.1007/978-0-387-46401-5_3

25 Abrahams S, Haylett W L, Johnson G, Carr J A, Bardien S. Antioxidant effects of curcumin in models of neurodegeneration, aging, oxidative and nitrosative stress: A review; 2019.

26. Salehi B, Stojanovic-Radic Z, Matejic J, Sharifi-Rad M, Anil K N, Martins N, et al. 2019 The therapeutic potential of curcumin: A review of clinical trials. Eur J Med Chem. 163, 527-545. doi:10.1016/j.ejmech.2018.12.016

14. Discussion, 2nd paragraph: please explain what is meant my ‘minority adipocytes with irreplaceable roles’. Also, in the discussion please elaborate on the evidence that supports the fact curcumin induced differentiation of adipocytes.

Response:

The phrase “minority adipocytes with irreplaceable roles” means brown adipocytes are small in quantity compared with white adipocytes, but the function of brown adipocytes should not be underestimated.

As for the statement of “curcumin induced differentiation of adipocytes”, we have revised it as follows:

“In the present study, the results of oil red O staining and triglyceride level in differentiated 3T3-L1 cells demonstrated that curcumin induced differentiation of the 3T3-L1 adipocytes and stimulated the lipid droplets intracellular accumulation.”

15. In the discussion, the authors suggest that precipitation of curcumin at high concentrations may account for the adverse effects on mitochondrial respiratory

function. This is a concern, as it raises questions about the validity of the data obtained at concentrations exceeding 10 μ M. In the methods and also here in the discussion the authors need to explain what measures they took to try and avoid precipitation of curcumin and also what evidence they have to support the suggestion – was there a visible precipitate for example? It would also be worth discussing how their chosen concentration range compares with the literature on in vitro curcumin experiments. Another possibility worth considering is a non-linear dose response; bell-shaped or U-shaped dose responses have been reported for various biological activities of other naturally occurring compounds – could this be the case for curcumin here? This should also be discussed.

Response:

We are sorry for the misinformation. Precipitation of curcumin is only a possible reason to explain the adverse effects on mitochondrial respiratory function. In fact, we did not observe the precipitate in curcumin solution at high concentrations under the microscope. We only guessed there might be some invisible precipitation. To avoid misunderstanding, we have deleted this sentence in the revision.

As for the basis on which we chose curcumin concentration range, we firstly tested the effect of various concentrations (10, 20, 35, and 50 μ M) of curcumin on cell viability of preadipocytes referring to the literatures, and then chose two low concentrations (10 and 20 μ M) for further experiments.

Therefore, we added statement in the method and discussion section as follows:

“The doses of curcumin used in this study was determined based on previous investigations [15, 16]”

“we firstly observed the metabolic effects of curcumin at different concentrations (10, 20, and 35 μ M) on the mitochondrial respiratory status of 3T3-L1 adipocytes referring to the literature [30]”

15. BB Aggarwal Y S S S. *The molecular targets and therapeutic uses of curcumin in health and disease: Springer Science & Business Media; 2007.*

-
16. Ferguson B S, Nam H, Morrison R F. 2016 Curcumin inhibits 3T3-L1 preadipocyte proliferation by mechanisms involving post-transcriptional p27 regulation. *Biochem Biophys Rep.* 5, 16-21. doi:10.1016/j.bbrep.2015.11.014
30. Lone J, Choi J H, Kim S W, Yun J W. 2016 Curcumin induces brown fat-like phenotype in 3T3-L1 and primary white adipocytes. *J Nutr Biochem.* 27, 193-202. doi:10.1016/j.jnutbio.2015.09.006

Thank you very much for the comment on reminding us of discussing a non-linear dose response, and we have revised this as follows:

“It was reported that some natural products and compounds showed a non-monotonic dose response - bell-curves for stimulatory biological effects [36], which may be one reason to explain our findings.”

36. Owen S C, Doak A K, Ganesh A N, Nedyalkova L, McLaughlin C K, Shoichet B K, et al. 2014 Colloidal drug formulations can explain "bell-shaped" concentration-response curves. *Acs Chem Biol.* 9, 777-784. doi:10.1021/cb4007584

16. *Discussion, page 13, line 55/56: The statement ‘directly or potential for further pathophysiological process’ doesn’t make sense. Please revise.*

Response:

Thank you very much. We have revised this part as follows:

“So far, it has been proved that various genes, such as those encoding GLUT-2 and IRS-2, target at PPAR γ directly and involve in extensive pathophysiological process, like glucose disposal [38]”

38. Kim H S, Hwang Y C, Koo S H, Park K S, Lee M S, Kim K W, et al. 2013 PPAR-gamma activation increases insulin secretion through the up-regulation of the free fatty acid receptor GPR40 in pancreatic beta-cells. *Plos One.* 8, e50128. doi:10.1371/journal.pone.0050128

17. Page 14, 1st line: please acknowledge that the PPAR γ antagonist did not completely inhibit the effects of curcumin on mitochondrial respiration, so other mechanisms may also be involved.

Response:

Thank you very much for your suggestion. We have revised this part as follows:

“The PPAR γ antagonist partially inhibited the ability of curcumin (10 μ M) to improve mitochondrial respiration, which indicates that PPAR γ may be the pivot point for further exploring the mechanisms of curcumin in adipocytes although other mechanisms may also be involved.”

18. Figure 5 legend: delete ‘DIO, diet-induced obesity’. This figure contains in vitro data so this abbreviation is not necessary.

Response:

Once again, we would like to thank you for all the insightful comments. We have revised this accordingly.

Reviewer: 2

This study describes the effect of curcumin on adipocyte metabolism in vitro and in vivo. Curcumin promoted adipogenesis in 3T3-L1 cells and affected mitochondrial respiration (low and high dose increased and decreased respiration, respectively). Low dose of curcumin increased expression of Ucp1 and genes related

to adipogenesis. In vivo, curcumin decreased lipid accumulation in both WAT and BAT and increased expression of Ucp1 and genes related to adipogenesis.

1. *Although the in vitro study was done properly, interpretation of the results concerning the implication of PPAR γ is misleading. I agree that the PPAR γ antagonist attenuated the effect of curcumin (figure 4), but most of the effects was still present. It suggests that most of the curcumin effects on respiration are PPAR γ -independent. Page 10 line 43, the interpretation should be that curcumin improves mitochondrial respiratory capacity of adipocyte in part through PPAR γ induction. The authors could measure more PPAR γ target gene expression (for instance, Angptl4, Cidea, Plin1) to make a stronger case on PPAR γ involvement.*

Response:

Thank you very much for your precise suggestion and we have revised the interpretation in our manuscript as suggested as follows:

“curcumin significantly improves the mitochondrial respiratory capacity of mature adipocytes in part through PPAR γ induction”

And we will try to measure more PPAR γ target gene expression to provide more evidence of PPAR γ involvement in the future work. We would like to thank you again for the insightful comment.

2. *I am concern that the dosage of curcumin could be toxic to the mice, which would explain the increase in lipolysis in WAT (in the previous publication) and increase FFA-induced browning in WAT. The authors should perform food intake experiments to demonstrate that the curcumin concentration does not have a detrimental effect on food consumption.*

Response:

We sincerely appreciate the reviewer's comments. On the one hand, curcumin is used as a food additive, which suggest it is a safe agent. On the other hand, phase I clinical trials have indicated that as high as 12 g of curcumin per day for over 3 months is well tolerated in humans [1].

Truly, toxicity studies of curcumin on mice or rats with prolonged high-dose intake are susceptible to hepatotoxicity [2], but the concentration we applied ($50 \text{ mg kg}^{-1} \text{ d}^{-1}$) is at a moderate level and has been found to be safe in most studies [3].

In addition, the effect of curcumin on food intake was observed, and the results showed that curcumin did not influence the food intake of obese mice (shown in the figure below).

[1] Lao, Christopher D., et al. "Dose escalation of a curcuminoid formulation." *BMC complementary and alternative medicine* 6.1 (2006): 1-4.

[2] Chainani-Wu, Nita. "Safety and anti-inflammatory activity of curcumin: a component of tumeric (*Curcuma longa*)." *The Journal of Alternative & Complementary Medicine* 9.1 (2003): 161-168.

[3] Sun, Xuejiao, et al. "Recent advances of curcumin in the prevention and treatment of renal fibrosis." *BioMed research international* 2017 (2017).

3. In this line, how the authors reconcile that the *in vitro* study showed that curcumin increase lipid accumulation while decreasing adipocyte size in the *in vivo* study?

Response:

Thank you very much for your comments.

Consistent with the literature [1], our in vitro study showed that curcumin significantly increased lipid accumulation, which indicates that curcumin promoted adipogenesis, from pre-adipocytes to mature adipocytes.

While the results of the in vivo studies showed that curcumin decreased mature adipocyte size because curcumin promoted white adipocytes browning and enhanced fuel utilization. In adipose tissues, curcumin enhance adipogenesis, so there are more adipocytes, but the cell volume became smaller.

These results are not contradictory as evidenced by the beneficial effect of curcumin on body weight gain and serum lipid profiles.

[1] Ariyanto E F, Nurazizah S I, Pamela Y, et al. *Effect of Curcuma longa Extract on Lipid Accumulation during Adipocyte Differentiation Using 3T3-L1 Cell Line*[J]. *Biomedical and Pharmacology Journal*, 2020, 13(3.): 1221-1226.

4. Page 14 line 12: it is not clear what “active factors” means.

Response: Thank you very much. we have modified it as follows:

“PPAR γ is a chief regulator that can promotes the differentiation process of white adipocytes collaborating with multiple other regulators, such as PGC-1, CCAAT/enhancer binding protein β (C/EBP β), phosphoenolpyruvate carboxykinase (PEPCK) and so on [41]”

41. Arner P. 2003 *The adipocyte in insulin resistance: Key molecules and the impact of the thiazolidinediones.* *Trends Endocrinol Metab.* 14, 137-145. doi:[http://dx.doi.org/10.1016/S1043-2760\(03\)00024-9](http://dx.doi.org/10.1016/S1043-2760(03)00024-9)

5. Page 14 line 19 and later: same for “Passageway”

Response:

UCP, a mitochondrial inner membrane protein, is a regulated proton channel or transporter [1]. Therefore, passageway means a channel, through which protons dissipate from the mitochondrial intermembrane space to the mitochondrial matrix.

[1] Gaudry MJ, Jastroch M (March 2019). "Molecular evolution of uncoupling proteins and implications for brain function". *Neuroscience Letters*. 696: 140–145